# VFIMamba: Video Frame Interpolation with State Space Models

**Guozhen Zhang**[1,2*]   **Chunxu Liu**[1]   **Yutao Cui**[2]   **Xiaotong Zhao**[2]   **Kai Ma**[2]   **Limin Wang**[1,3†]

[1]State Key Laboratory for Novel Software Technology, Nanjing University
[2]Platform and Content Group (PCG), Tencent    [3]Shanghai AI Lab

https://github.com/MCG-NJU/VFIMamba

## Abstract

Inter-frame modeling is pivotal in generating intermediate frames for video frame interpolation (VFI). Current approaches predominantly rely on convolution or attention-based models, which often either lack sufficient receptive fields or entail significant computational overheads. Recently, Selective State Space Models (S6) have emerged, tailored specifically for long sequence modeling, offering both linear complexity and data-dependent modeling capabilities. In this paper, we propose VFIMamba, a novel frame interpolation method for efficient and dynamic inter-frame modeling by harnessing the S6 model. Our approach introduces the Mixed-SSM Block (MSB), which initially rearranges tokens from adjacent frames in an interleaved fashion and subsequently applies multi-directional S6 modeling. This design facilitates the efficient transmission of information across frames while upholding linear complexity. Furthermore, we introduce a novel curriculum learning strategy that progressively cultivates proficiency in modeling inter-frame dynamics across varying motion magnitudes, fully unleashing the potential of the S6 model. Experimental findings showcase that our method attains state-of-the-art performance across diverse benchmarks, particularly excelling in high-resolution scenarios. In particular, on the X-TEST dataset, VFIMamba demonstrates a noteworthy improvement of **0.80** dB for 4K frames and **0.96** dB for 2K frames.

## 1 Introduction

Video Frame Interpolation (VFI), a fundamental task in video data processing, is gaining substantial attention for its ability to generate intermediate frames between consecutive frames (Liu et al., 2017). Its utility spans many practical applications, including creating slow-motion videos through temporal upsampling (Jiang et al., 2018), enhancing video refresh rates (Reda et al., 2022), and generating novel views (Flynn et al., 2016; Szeliski, 1999). VFI typically encompass two primary stages (Zhang et al., 2023): firstly, conducting the inter-frame modeling of input consecutive frames; and secondly, leveraging the acquired information to estimate inter-frame motion and generate intermediate frame appearance. In practice, VFI often deals with high-resolution inputs (e.g., 4K) (Sim et al., 2021), which results in significant object displacement and imposes high demands on the large receptive field of the modules that model information between frames. Additionally, since VFI is commonly applied to long-duration videos such as movies, model speed is also of paramount importance. Thus, *striking a delicate balance between a sufficient receptive field and fast processing speed in modeling inter-frame information* is the key in crafting effective VFI models.

---

[*]Work is done during internship at Tencent PCG. [†]Corresponding author (lmwang@nju.edu.cn).

38th Conference on Neural Information Processing Systems (NeurIPS 2024).

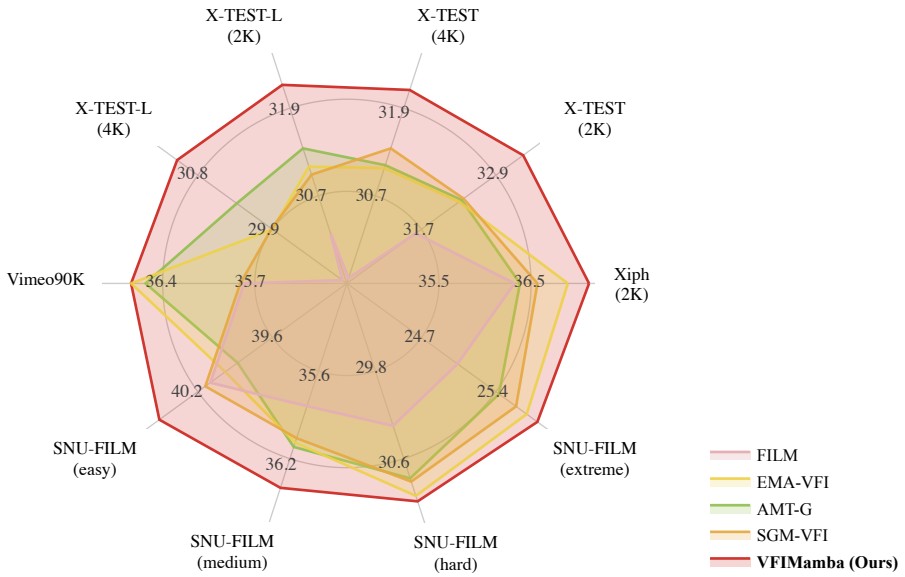

Figure 1: Equipped with the S6 model, our VFIMamba achieves the state-of-the-art performance on benchmarks across different input resolutions.

Current methods for modeling inter-frame information predominantly rely on convolutional neural networks (CNNs) (Liu et al., 2017; Kong et al., 2022; Huang et al., 2022) and attention-based models (Lu et al., 2022a; Zhang et al., 2023; Park et al., 2023; Liu et al., 2024a). However, as illustrated in the first three rows of Table 1, these methods either (1) lack flexibility and cannot adaptively model based on the input, (2) do not have sufficient receptive fields to capture inter-frame correlations at high resolutions, or (3) suffer from prohibitive computational complexity.

On the other hand, Natural Language Processing (NLP) has recently witnessed the emergence of structured state space models (SSMs) (Gu et al., 2021). Theoretically, SSMs combine the benefits of Recurrent Neural Networks (RNNs) and CNNs, leveraging the global receptive field characteristic of RNNs and the computational efficiency inherent in CNNs. One particularly notable SSM is the Selective State Space Model (S6), also known as Mamba (Gu & Dao, 2023), which has garnered significant attention within the vision community. Mamba's novel feature of making SSM parameters time-variant (i.e., data-dependent) enables it to effectively select relevant context within sequences, a crucial factor for enhancing model performance. However, to the best of our knowledge, *S6 has not yet been applied to low-level video tasks*.

To address the challenges faced by current VFI models and to explore the potential of the S6 model (Gu & Dao, 2023) in low-level video tasks, we propose VFIMamba, a novel frame interpolation method that adapts the S6 model for efficient and dynamic inter-frame modeling. As shown in Table 1, VFIMamba provides the advantages of a global receptive field with linear complexity while maintaining data-dependent adaptability.

Specifically, we introduce the Mixed-SSM Block (MSB) to replace existing modules for inter-frame information transfer. The original S6 model can only process a single sequence, so it is necessary to merge tokens from two frames into one sequence for effective inter-frame modeling. After thorough analysis and exploration, we figured out that interleaving tokens from both frames into a "super image" is more suitable for VFI. We then conduct multi-directional SSMs on this image to model inter-frame information. This interleaved approach facilitates interactions between adjacent tokens from different frames during sequence modeling and ensures that the intermediate tokens of any pair of tokens in the sequence are from their spatiotemporal neighborhood. By stacking MSB modules, our model effectively handles complex inter-frame information exchange. Finally, we use the extracted inter-frame features to estimate motion and generate the appearance of intermediate frames.

While the S6 model boasts the advantages listed in Table 1, it is crucial to employ appropriate training strategies to fully exploit its potential. Inspired by Bengio et al. (2009), we propose a novel curriculum learning strategy that progressively teaches the model to handle inter-frame modeling across varying motion amplitudes. Specifically, while maintaining training on Vimeo90K (Xue et al.,

Table 1: Comparison of the model design for inter-frame modeling of VFIMamba and existing methods. VFIMamba enjoys both the advantages of a large receptive field and linear complexity.

| Model | Data-dependent | Linear complexity | Global receptive field | Representative method |
|---|---|---|---|---|
| CNN | ✗ | ✓ | ✗ | RIFE (Huang et al., 2022) |
| Attention | ✓ | ✗ | ✓ | SGM-VFI (Liu et al., 2024a) |
| Local Attention | ✓ | ✓ | ✗ | EMA-VFI (Zhang et al., 2023) |
| Mamba | ✓ | ✓ | ✓ | VFIMamba (our work) |

2019), we incrementally introduce large motion data from X-TRAIN (Sim et al., 2021), increasing the motion amplitude over time. This learning strategy enables VFIMamba to perform well across a wide range of motion amplitudes, thereby fully unleashing the potential of the S6 model.

To validate the effectiveness of VFIMamba across various types of video data, we conduct extensive testing on different benchmarks. As shown in Figure 1, VFIMamba achieves the state-of-the-art (SOTA) performance across diverse datasets. This is particularly evident in high-resolution and large-motion datasets such as X-TEST (Sim et al., 2021) and SNU-FILM (Choi et al., 2020).

**Contribution.** In summary, the contributions of this paper are as follows: (1) We are the first to adapt the S6 model to the video frame interpolation task. To better adapt the model for this task, we introduce the Mixed-SSM Block (MSB), providing a solid foundation for future architectural exploration in frame interpolation. (2) We propose a novel curriculum learning strategy that incrementally introduces data with varying motion amplitudes, thereby fully harnessing the potential of the S6 model. (3) Our model achieves the state-of-the-art performance across a wide range of datasets, potentially sparking interest in the exploration of the S6 model within the video low-level community.

## 2 Related work

### 2.1 Video frame interpolation

The performance of VFI methods has seen significant advancements with the emergence of deep learning models. **(1)** CNNs-based approaches (Bao et al., 2019; Liu et al., 2017; Huang et al., 2022; Niklaus & Liu, 2018; Choi et al., 2020; Zhu et al., 2024b; Jia et al., 2022; Niklaus et al., 2017; Kalluri et al., 2023): Initially, DVF (Liu et al., 2017) utilized a U-Net-like (Ronneberger et al., 2015) network to model two input frames and predicted the voxel flow for warping the two frames into the intermediate frame. Following this, CtxSyn (Niklaus & Liu, 2018) introduced ContextNet and RefineNet, where ContextNet extracts context information from each frame, and RefineNet further refines the coarse intermediate frame produced by warping. RIFE (Huang et al., 2022) proposed a novel, efficient framework that employs self-distillation to significantly reduce computational load and parameters while maintaining high performance. Due to its simplicity, many convolutional modeling works (Kong et al., 2022; Jia et al., 2022) have improved upon RIFE. **(2)** Attention-based approaches (Lu et al., 2022a; Zhang et al., 2023; Park et al., 2023; Liu et al., 2024a): VFIFormer (Lu et al., 2022a) was the first to use attention to model inter-frame information, replacing the encoder part of U-Net with Transformer blocks. After that, EMA-VFI (Zhang et al., 2023) uses Swin-based (Liu et al., 2021) local attention to simultaneously capture local appearance and motion information. AMT (Li et al., 2023) used a multi-scale cost-volume construction similar to RAFT (Teed & Deng, 2020) to further enhance motion modeling capabilities. BiFormer (Park et al., 2023) introduced quasi-global bilateral attention to further increase the receptive field for large motions. SGM-VFI (Liu et al., 2024a) introduced sparse global matching to model motion between frames. However, current models struggle to balance sufficient receptive fields with computational overhead. In contrast, our method introduces the first interpolation model based on State Space Models (SSMs) (Gu & Dao, 2023) and further pushes the performance boundaries of VFI tasks.

### 2.2 State space models

In the field of NLP, SSMs (Gu et al., 2021; Smith et al., 2022; Mehta et al., 2022; Fu et al., 2022) have recently emerged as one of the most promising contenders to challenge the dominance of Transformers. The Structured State Space Sequence Model (S4) (Gu et al., 2021) was initially introduced for linear complexity modeling of long sequences. Subsequent works have improved

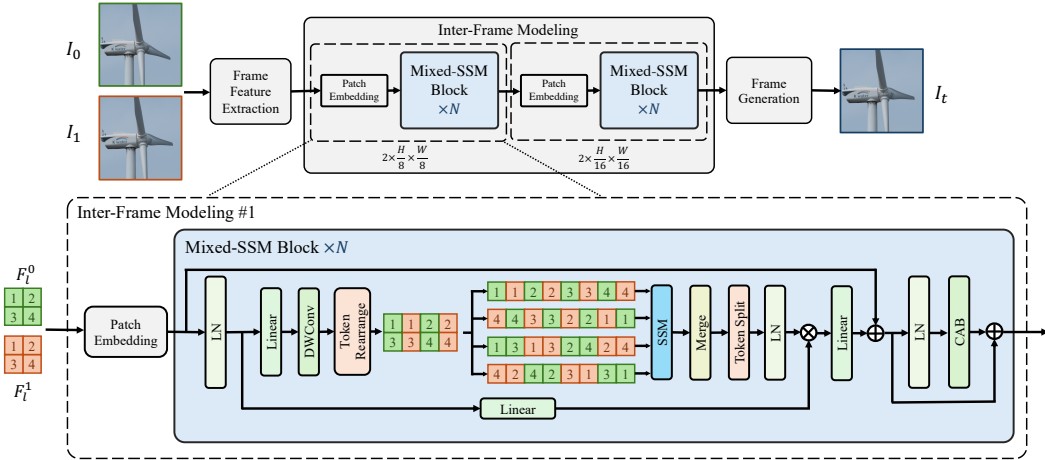

Figure 2: Overall pipeline of VFIMamba. Firstly, a lightweight feature extractor is employed to encode the input frames into shallow features. Subsequently, we utilize the Mixed-SSM Block (MSB) to conduct inter-frame modeling using S6, iterating $N$ times at each scale. Finally, these inter-frame features are leveraged to generate the intermediate frame.

its computational efficiency and model capacity. S5 (Smith et al., 2022) proposed a parallel scan and MIMO SSM, and GSS (Mehta et al., 2022) enhanced the model's capability by introducing gated mechanisms. Mamba (S6) (Gu & Dao, 2023) has recently stood out due to its data-dependent parameter generation and efficient hardware implementation, outperforming Transformers in long-sequence NLP tasks. In the visual domain, Vim (Zhu et al., 2024a) was the first to permute 2D images into sequences for global modeling using bidirectional SSMs. Vmamba (Liu et al., 2024b) extended to four directions and introduced a hierarchical structural design. VideoMamba (Li et al., 2024) was the first to apply S6 in the video domain by permuting all frames into a spatiotemporal sequence. MambaIR (Guo et al., 2024) was the first to use the S6 model for image restoration tasks, achieving superior performance over Transformers. In this work, we explore the potential of the S6 model in VFI tasks, validating its effectiveness through detailed analysis and experimentation.

## 3 Method

### 3.1 Preliminaries

SSMs are mainly inspired by the continuous linear time-invariant (LTI) systems, which apply an implicit latent state $h(t) \in \mathbb{R}^N$ to map a 1-dimensional sequence or function $x(t) \in \mathbb{R} \rightarrow y(t) \in \mathbb{R}$. Specifically, SSMs can be formulated as an ordinary differential equation (ODE):

$$h'(t) = Ah(t) + Bx(t), \tag{1}$$
$$y(t) = Ch(t), \tag{2}$$

where contains evolution matrix $A \in \mathbb{R}^{N \times N}$, projection parameters $B \in \mathbb{R}^{N \times 1}$ and $C \in \mathbb{R}^{1 \times N}$. However, it is hard to solve the above differential equation in deep learning settings and needs to be approximated through discretization. Recent SSMs (Gu et al., 2021) propose to introduce a timescale parameter $\Delta$ to transform the $A, B$ to their discrete counterparts $\bar{A}, \bar{B}$, i.e.,

$$h_t = \bar{A}h_{t-1} + \bar{B}x_t, \tag{3}$$
$$y_t = Ch_t, \tag{4}$$
$$\bar{A} = \exp(\Delta A), \tag{5}$$
$$\bar{B} = (\Delta A)^{-1}(\exp(\Delta A - I)) \cdot \Delta B. \tag{6}$$

The above SSMs are performed for each channel separately and their parameters are data-independent, meaning that $\bar{A}, \bar{B}$ and $C$ are the same for any input in the same channel, limiting their flexibility in sequence modeling. Mamba (Gu & Dao, 2023) proposes the selective SSMs (S6), which dynamically

generate the parameters for each input data $x_i \in \mathbb{R}^L$ using the entire $x_i$:

$$B_i = S_B x_i, \qquad C_i = S_C x_i, \qquad \Delta_i = \texttt{Softplus}\,(S_\Delta x_i), \qquad (7)$$

where $S_B \in \mathbb{R}^{N \times L}, S_C \in \mathbb{R}^{N \times L}, S_\Delta \in \mathbb{R}^{L \times L}$ are linear projection layers. The $B_i$ and $C_i$ are shared for all channels of $x_i$, $\Delta_i$ contains $\Delta$ of $L$ channels, and $A$ are the same as previous SSMs. By the discretization in equations 5 and 6, $\bar{A}$ and $\bar{B}$ become different based on input data.

## 3.2 Overall pipeline

Given two consecutive frames $I_0, I_1 \in \mathbb{R}^{H \times W \times 3}$ along with a timestep $t$, the objective of the frame interpolation task is to generate the intermediate frame $I_t \in \mathbb{R}^{H \times W \times 3}$. As illustrated in Figure 2, the overall pipeline of VFIMamba consists of three main components: frame feature extraction, inter-frame modeling, and frame generation. Firstly, we employ a set of lightweight convolutional layers to independently extract shallow features from each frame, progressively reducing the resolution to facilitate more efficient inter-frame modeling. This process can be formulated as:

$$F_l^i = \mathcal{L}(I_i), \qquad (8)$$

where $\mathcal{L}$ represents the set of convolutional layers, and $F_l^i$ denotes the extracted low-level feature for $I_i$. Next, we perform multi-resolution inter-frame modeling using the proposed Mixed-SSM Block (MSB). Each scale comprises $N$ MSBs, and downsampling between scales is achieved through overlapping patch embedding (Wang et al., 2022). We define the resulting inter-frame features as $F_{ssm}^i$. Finally, we utilize these high-quality inter-frame features for frame generation, which involves motion estimation between two frames and appearance refinement:

$$I_t = \mathcal{G}(F_{ssm}^0, F_{ssm}^1), \qquad (9)$$

where $\mathcal{G}$ denotes the frame generation network. Since this work primarily focuses on exploring the use of SSMs for inter-frame modeling, we largely follow the design from Zhang et al. (2023) and Huang et al. (2022) for the frame generation components, with detailed specifications provided in the appendix.

## 3.3 State space models for inter-frame modeling

Effective inter-frame modeling is crucial for frame interpolation tasks (Zhang et al., 2023). Methods such as RIFE (Huang et al., 2022) and EMA-VFI (Zhang et al., 2023) employ simple convolution layers or local attention for inter-frame modeling, achieving high inference speeds but limiting receptive field. Conversely, SGM-VFI (Liu et al., 2024a) uses global inter-frame attention for motion estimation, which improves performance but sacrifices efficiency. In this work, we propose to use state space models (SSMs), specifically S6 (Gu & Dao, 2023), to achieve both efficiency and effectiveness in inter-frame modeling.

### 3.3.1 Mixed-SSM block

To facilitate more efficient inter-frame information exchange globally, we utilize SSMs for inter-frame modeling. As illustrated in Figure 2, we introduce the Mixed-SSM Block (MSB) for integrate the S6 model into VFI frameworks. The overall design of the MSB is analogous to Transformer (Vaswani et al., 2017) blocks, but with two pivotal distinctions: (1) We substitute the attention mechanism with an enhanced S6 Block (Gu & Dao, 2023), which could conduct global inter-frame modeling with linear complexity. (2) Drawing inspiration from Guo et al. (2024) and Behrouz et al. (2024), which identified the lack of locality and inter-channel interaction in SSMs, we replace the multilayer perceptron (MLP) with a Channel-Attention Block (CAB) (Hu et al., 2018).

The original S6 model is limited to processing one-dimensional sequences, necessitating a strategy for scanning the feature maps of two input frames for inter-frame modeling. As depicted in Figure 3, there are two primary methods to rearrange the two frames: **sequential rearrangement**, where the frames are concatenated into a single super image, and **interleaved rearrangement**, where the tokens of the two frames are interleaved to form a super image. Regardless of the rearrangement method, following Liu et al. (2024b), the super image can be scanned in four directions: horizontally, vertically, and in their respective reverse directions. The S6 Block is then employed to model each direction independently, and the resulting sequences are rearranged and merged back.

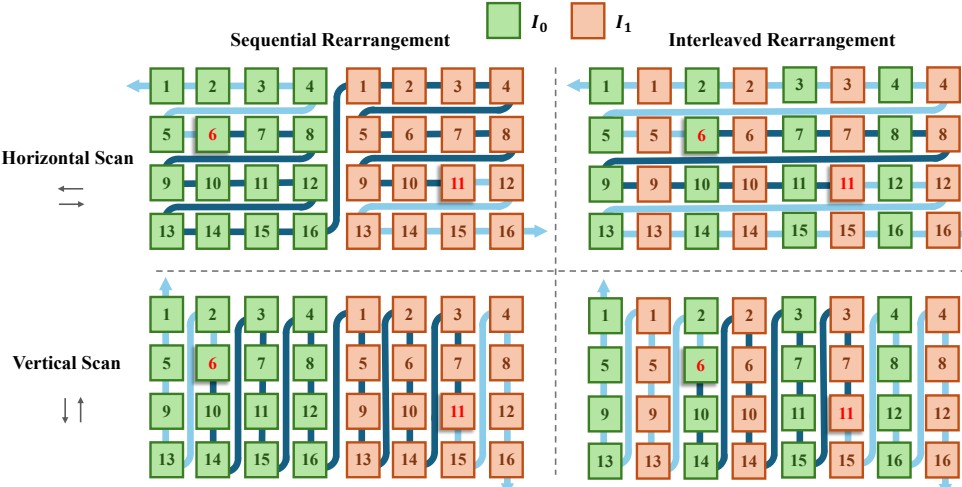

Figure 3: Visualizations of different rearrangement methods and scan directions. The choice of rearrangement strategy impacts the information flow during inter-frame modeling with S6. For example, consider the 6-th token from $I_0$ and the 11-th token from $I_1$. In sequential rearrangement, the intermediate tokens introduce too many irrelevant tokens, whereas interleaved rearrangement more effectively preserves the spatiotemporal locality.

### 3.3.2   Analysis on rearrangement strategies

Here, we discuss which rearrangement method is better for inter-frame modeling in the context of frame interpolation. First, let us introduce a conclusion from (Ali et al., 2024): the S6 layers can be approximated as hidden attention layers, with the attention weights given by:

$$\alpha_{i,j} \approx Q_i K_j H_{i,j}, \tag{10}$$

where

$$Q_i = S_C x_i, \quad K_j = \text{ReLU}\left(S_\Delta x_j\right) S_B x_j, \quad H_{i,j} = \exp(\sum_{\substack{k \in (i,j) \\ S_\Delta x_k > 0}} (S_\Delta x_k))A, \tag{11}$$

In this formulation, $\alpha_{i,j}$ represents the hidden attention weight of the $j$-th token $x_j$ to the $i$-th token $x_i$ in the sequence. Unlike attention, which calculates weights based solely on the information from tokens $x_i$ ($Q_i$) and $x_j$ ($K_j$), the S6 model includes $H_{i,j}$, which encompasses the contextual information between the $i$-th and $j$-th tokens in the sequence. Based on this conclusion, we observe that in the interleaved rearrangement, the intermediate tokens of any pair of tokens in the sequence are from their spatiotemporal neighborhood. This means that $H_{i,j}$ incorporates more local modeling, which is beneficial for low-level tasks like frame interpolation. Additionally, the number of intermediate tokens between spatiotemporally adjacent tokens is generally smaller in the interleaved rearrangement. In contrast, in the sequential rearrangement, even spatiotemporally adjacent tokens are separated by many unrelated tokens in the sequence. This can introduce noise and interfere with the modeling of the relationship between these tokens. A specific example can be seen in Figure 3, where the tokens between the 6-th token of the first frame and the 11-th token of the second frame differs significantly between the two rearrangement methods. In summary, we believe that for video frame interpolation, the interleaved rearrangement method is more suitable for better local spatially-aware processing. Our experiments, detailed in Section 4.2, further validate this conclusion.

### 3.4   Curriculum learning for VFIMamba

Despite the advantageous characteristics of the S6 model, such as data dependence and global receptive field, it is crucial to fully exploit its potential through appropriate training strategies. Currently, two main training strategies are employed for frame interpolation: (1) **Vimeo90K Only**: Most methods training models exclusively on the Vimeo90K (Xue et al., 2019). Although Vimeo90K offers a rich variety of video content, as analyzed by Liu et al. (2024a) and Kiefhaber et al. (2024),

Table 2: Quantitative comparison with SOTA methods on the low-resolution datasets, in terms of PSNR/SSIM (Wang et al., 2004). The best results and the second best results are **boldfaced** and underlined respectively. FLOPs was calculated for 720p input. ★ indicates the results copied from Zhang et al. (2023) and Li et al. (2023). In "Training Dataset", "V" stands for the triplet of Vimeo-90K and "X" stands for X-TRAIN. In "Runtime", we evaluate the inference speed of each method on $1024 \times 1024$ resolution inputs by a 2080Ti GPU. In "Average", we calculate the average performance of each method in terms of PSNR and SSIM.

| | Training Dataset | Vimeo-90K (Xue et al., 2019) | UCF101 (Soomro et al., 2012) | SNU-FILM (Choi et al., 2020) | | | | Average | FLOPs (T) | Runtime (ms) |
| --- | --- | --- | --- | --- | --- | --- | --- | --- | --- | --- |
| | | | | easy | medium | hard | extreme | | | |
| DAIN★ (Bao et al., 2019) | V | 34.71/0.9756 | 34.99/0.9683 | 39.73/0.9902 | 35.46/0.9780 | 30.17/0.9335 | 25.09/0.8584 | 33.36/0.9507 | 5.51 | 897.8 |
| AdaCof★ (Lee et al., 2020) | V | 34.47/0.9730 | 34.90/0.9680 | 39.80/0.9900 | 35.05/0.9754 | 29.46/0.9244 | 24.31/0.8439 | 33.00/0.9458 | 0.36 | 85.1 |
| CAIN★ (Choi et al., 2020) | V | 34.65/0.9730 | 34.91/0.9690 | 39.89/0.9900 | 35.61/0.9776 | 29.90/0.9292 | 24.78/0.8507 | 33.29/0.9483 | 1.29 | 102.4 |
| Softsplat (Niklaus & Liu, 2020) | V | 36.13/0.9805 | 35.39/0.9697 | 40.26/0.9911 | 36.07/0.9798 | 30.53/0.9365 | 25.16/0.8604 | 33.92/0.9530 | 0.94 | 266.4 |
| XVFI (Sim et al., 2021) | V | 35.09/0.9759 | 35.17/0.9685 | 39.93/0.9907 | 35.37/0.9782 | 29.58/0.9276 | 24.17/0.8450 | 33.22/0.9477 | 0.37 | 165.2 |
| M2M-VFI (Hu et al., 2022) | V | 35.47/0.9778 | 35.28/0.9694 | 39.66/0.9904 | 35.74/0.9794 | 30.30/0.9360 | 25.08/0.8604 | 33.59/0.9522 | 0.26 | 60.9 |
| RIFE (Huang et al., 2022) | V | 35.61/0.9779 | 35.28/0.9690 | 39.80/0.9903 | 35.76/0.9787 | 30.36/0.9351 | 25.27/0.8601 | 33.68/0.9519 | 0.20 | 35.2 |
| IFRNet-L (Kong et al., 2022) | V | 36.20/0.9808 | 35.42/0.9698 | 40.10/0.9906 | 36.12/0.9797 | 30.63/0.9368 | 25.26/0.8609 | 33.96/0.9531 | 0.79 | 115.3 |
| EMA-VFI-S (Zhang et al., 2023) | V | 36.07/0.9797 | 35.34/0.9696 | 39.81/0.9906 | 35.88/0.9795 | 30.69/0.9375 | 25.47/0.8632 | 33.88/0.9534 | 0.20 | 76.4 |
| EMA-VFI (Zhang et al., 2023) | V | **36.64**/**0.9819** | **35.48**/0.9701 | 39.98/0.9910 | 36.09/**0.9801** | 30.94/0.9392 | 25.69/0.8661 | 34.14/0.9547 | 0.91 | 239.6 |
| AMT-L (Li et al., 2023) | V | 36.35/0.9815 | 35.39/0.9698 | 39.95/**0.9913** | 36.09/**0.9805** | 30.75/0.9384 | 25.41/0.8638 | 33.99/0.9542 | 0.58 | 183.42 |
| AMT-G (Li et al., 2023) | V | 36.53/0.9817 | 35.41/0.9699 | 39.88/**0.9913** | 36.12/**0.9805** | 30.78/0.9385 | 25.43/0.8644 | 34.03/0.9544 | 2.07 | 403.7 |
| SGM-VFI (Liu et al., 2024a) | V+X | 35.81/0.9793 | 35.34/0.9693 | 40.14/0.9907 | 36.06/0.9795 | 30.81/0.9375 | 25.59/0.8646 | 33.96/0.9535 | 1.78 | 942.9 |
| VFIMamba-S | V+X | 36.09/0.9800 | 35.36/0.9696 | 40.21/0.9909 | 36.17/0.9800 | 30.80/0.9381 | 25.59/0.8655 | 34.04/0.9540 | 0.24 | 128.0 |
| VFIMamba | V+X | **36.64**/**0.9819** | 35.45/**0.9702** | **40.51**/0.9912 | **36.40**/**0.9805** | **30.99**/**0.9401** | **25.79**/**0.8682** | **34.30**/**0.9554** | 0.94 | 310.9 |

Table 3: Quantitative comparison with SOTA methods on high-resolution datasets. "OOM" indicates "Out of Memory" on V100. All results are obtained through the same evaluation procedure.

| | Training Dataset | X-TEST (Sim et al., 2021) | | X-TEST-L (Liu et al., 2024a) | | Xiph (Montgomery, 1994) | | Average |
| --- | --- | --- | --- | --- | --- | --- | --- | --- |
| | | 2K | 4K | 2K | 4K | 2K | 4K | |
| XVFI (Sim et al., 2021) | X | 31.15/0.9144 | 30.12/0.9045 | 29.82/0.8951 | 29.02/0.8866 | 34.76/0.9258 | 32.84/0.8810 | 31.29/0.9012 |
| M2M-VFI (Hu et al., 2022) | V | 32.13/0.9258 | 30.89/0.9138 | 30.90/0.9092 | 29.73/0.9001 | 36.44/0.9427 | 33.92/0.8992 | 32.34/0.9151 |
| RIFE (Huang et al., 2022) | V | 31.10/0.8972 | 30.13/0.8927 | 29.87/0.8805 | 28.98/0.8756 | 36.19/0.9380 | 33.76/0.8940 | 31.67/0.8963 |
| FILM (Reda et al., 2022) | V | 31.61/0.9174 | OOM | 30.18/0.8960 | OOM | 36.32/0.9343 | 33.27/0.8760 | / |
| IFRNet-L (Kong et al., 2022) | V | 31.78/0.9147 | 30.66/0.9050 | 30.76/0.8963 | 29.74/0.8884 | 36.21/0.9374 | 34.25/0.8946 | 32.23/0.9061 |
| FLDR (Nottebaum et al., 2022) | X | 31.12/0.9092 | 30.46/0.9041 | 29.90/0.8906 | 29.30/0.8879 | 34.80/0.9280 | 33.00/0.8862 | 31.43/0.9010 |
| BiFormer (Park et al., 2023) | V+X | 31.32/0.9200 | 31.32/0.9215 | 30.36/0.9068 | 30.14/0.9069 | 34.20/0.9246 | 33.49/0.8953 | 31.81/0.9125 |
| EMA-VFI-S (Zhang et al., 2023) | V | 30.91/0.9000 | 29.91/0.8951 | 29.51/0.8775 | 28.60/0.8733 | 36.55/0.9421 | 34.25/0.9020 | 31.62/0.8983 |
| AMT-L (Li et al., 2023) | V | 32.08/0.9277 | 30.96/0.9147 | 31.09/0.9103 | 30.12/0.9019 | 36.27/0.9402 | 34.49/0.9030 | 32.50/0.9163 |
| AMT-G (Li et al., 2023) | V | 32.35/0.9300 | 31.12/0.9157 | 31.35/0.9125 | 30.33/0.9036 | 36.38/0.9410 | **34.63**/0.9039 | 32.69/0.9178 |
| SGM-VFI (Liu et al., 2024a) | V+X | 32.38/0.9272 | 31.35/0.9179 | 30.99/0.9072 | 29.91/0.8972 | 36.57/0.9424 | 34.23/0.9021 | 32.57/0.9157 |
| VFIMamba-S | V+X | 32.84/0.9328 | 31.73/0.9238 | 31.58/0.9169 | 30.50/0.9077 | 36.72/0.9428 | 34.32/0.9034 | 32.95/0.9212 |
| VFIMamba | V+X | **33.34**/**0.9361** | **32.15**/**0.9246** | **32.22**/**0.9259** | **31.05**/**0.9159** | **37.13**/0.9451 | 34.62/**0.9059** | **33.42**/**0.9256** |

its motions have limited magnitude. This restriction hampers the model's performance on inputs with large motions or high resolution. (2) **Sequential Learning**: To mitigate the limitations of training solely on Vimeo90K, some methods (Liu et al., 2024a; Park et al., 2023) further train the model on X-TRAIN (Sim et al., 2021), a dataset characterized by large motions and high-resolution content, after initial training on Vimeo90K. While this approach successfully enhances the model's performance on high-resolution data, it often leads to the forgetting of the small-motion modeling capabilities acquired from Vimeo90K.

To address these issues and fully exploit the potential of the S6 model, inspired by Bengio et al. (2009), we propose a **curriculum learning** strategy for learning inter-frame modeling capabilities across varying motion magnitudes while maintaining the ability to model small motions. Specifically, while continuing training on Vimeo90K, we progressively incorporated data from X-TRAIN. The original size of X-TRAIN is $512 \times 512$, to co-train with Vimeo90K, we first resize the frames to $S \times S$ and then random crop to the same as Vimeo90K. Every $T$ epochs, the resized size $S$ is increased by 10% (starting from 256), and the temporal interval between selected frames is doubled (starting from 2), which means the motion magnitude increases as training progresses. This strategy enables the model to gradually learn inter-frame modeling capabilities across varying motion magnitudes, starting with smaller motions and progressing to larger ones.

## 4 Experiments

We provide two models: a lightweight model, VFIMamba-S, and a high-performance model, VFI-Mamba. Both models have $N = 3$; the only difference is that VFIMamba has twice the number of channels as VFIMamba-S. As described in Section 3.4, we employ a curriculum learning strategy in which $T = 50$ and train for 300 epochs in total. More model configurations and training details are provided in the appendix.

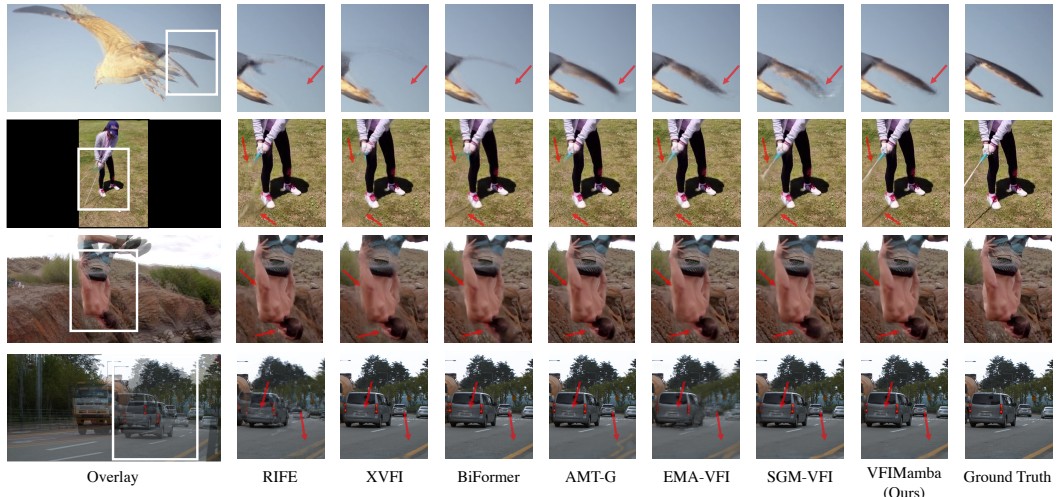

| Overlay | RIFE | XVFI | BiFormer | AMT-G | EMA-VFI | SGM-VFI | VFIMamba (Ours) | Ground Truth |

Figure 4: Visualizations from SNU-FILM (Reda et al., 2022) and X-TEST (Sim et al., 2021).

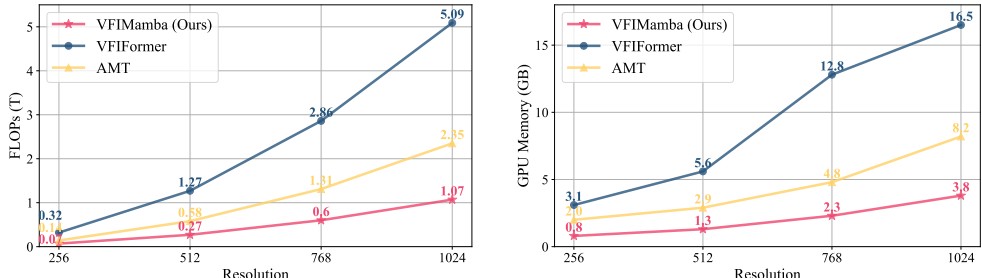

Figure 5: Comparisons of FLOPs and GPU memory usage with increasing resolution input.

## 4.1 Comparison with the State-of-the-Art Methods

**Quantitative comparison.** To validate the versatility of our proposed VFIMamba, we evaluated its performance (PSNR/SSIM) (Wang et al., 2004) across a variety of well-known benchmarks with different resolutions. The low-resolution datasets include Vimeo90K ($448 \times 256$) (Xue et al., 2019), UCF101 ($256 \times 256$) (Soomro et al., 2012), and SNU-FILM ($1280 \times 720$) (Reda et al., 2022). Notably, SNU-FILM is categorized into four levels of difficulty based on frame intervals: Easy, Medium, Hard, and Extreme. The high-resolution datasets include X-TEST (Sim et al., 2021), X-TEST-L (a more challenging subset selected by Liu et al. (2024a)), and Xiph (Montgomery, 1994). Originally, these datasets are in 4K resolution, and following Zhang et al. (2023), we also resize them to 2K for testing.

For 8x interpolation, we followed the testing procedure of FILM (Reda et al., 2022) and used an iterative approach for frame interpolation. Specifically, we first generated an intermediate frame based on the input two frames, and then, using a divide-and-conquer strategy, we further divided the first frame and the generated intermediate frame, as well as the generated intermediate frame and the last frame, to iteratively generate the remaining frames.

As shown in Tables 2 and 3, VFIMamba achieves state-of-the-art performance on almost all datasets with FLOPs comparable to efficient models (Kong et al., 2022; Zhang et al., 2023). Specifically, in large motion scenarios like X-TEST and X-TEST-L, VFIMamba demonstrates a noteworthy improvement compared with previous metod. This excellent performance underscores the potential of the S6 model in frame interpolation tasks, and we hope it will draw more attention to the application of SSMs in low-level video tasks.

**Qualitative comparison.** To further validate the practical effectiveness of VFIMamba, we also present a visual comparison with other frame interpolation methods. As illustrated in Figure 4, the arrows highlight areas where our method excels. VFIMamba demonstrates superior motion estimation

Table 4: Ablation on different models for inter-frame modeling. We use the V100 GPU for evaluating and "OOM" indicates "Out of Memory".

| Model | Vimeo90K | X-TEST | | SNU-FILM | | Params (M) | 720p Inference Time (ms) |
| --- | --- | --- | --- | --- | --- | --- | --- |
| | | 2K | 4K | hard | extreme | | |
| w/o S6 | 35.62/0.9771 | 28.94/0.8517 | 27.12/0.8436 | 30.41/0.9341 | 25.14/0.8567 | 16.1 | **51** |
| Convolution | 35.86/0.9790 | 31.58/0.9167 | 30.24/0.9044 | 30.61/0.9365 | 25.49/0.8631 | 23.4 | 55 |
| Local Attention | 35.92/0.9790 | 30.49/0.8917 | 30.00/0.8845 | 30.47/0.9338 | 25.46/0.8625 | **15.6** | 59 |
| Full Attention | 36.04/0.9798 | OOM | OOM | 30.55/0.9367 | 25.35/0.8602 | **15.6** | 336 |
| S6 | **36.12/0.9802** | **32.84/0.9328** | **31.73/0.9238** | **30.80/0.9381** | **25.59/0.8655** | 16.8 | 77 |

Table 5: Ablation on different rearrangement approachs. "Sequential" means sequential rearrangement and "Interleaved" represents interleaved rearrangement .

| Horizontal Scan | Vertical Scan | Vimeo-90K | X-TEST | | SNU-FILM | |
| --- | --- | --- | --- | --- | --- | --- |
| | | | 2K | 4K | hard | extreme |
| Sequential | Sequential | 35.55/0.9765 | 28.07/0.8327 | 26.75/0.8327 | 30.24/0.9319 | 25.03/0.8545 |
| Sequential | Interleaved | 35.76/0.9784 | 31.69/0.9226 | 30.45/0.9078 | 30.32/0.9342 | 25.21/0.8611 |
| Interleaved | Sequential | 35.79/0.9785 | 31.49/0.9221 | 30.35/0.9053 | 30.12/0.9331 | 25.11/0.8602 |
| Interleaved | Interleaved | **36.12/0.9802** | **32.84/0.9328** | **31.73/0.9238** | **30.80/0.9381** | **25.59/0.8655** |

and detail preservation in high-motion scenarios compared to other methods. This further substantiates that the incorporation of the S6 model enhances the performance of inter-frame interpolation tasks.

**Efficiency comparison.** To validate the efficiency of VFIMamba, we compared the FLOPs and GPU memory usage required by various high-performance methods (Li et al. (2023) and Lu et al. (2022b)) as the resolution increases. As shown in Figure 5, VFIMamba requires significantly fewer FLOPs and GPU memory as the input resolution grows, demonstrating the effectiveness of the S6 model in the VFI task.

## 4.2 Ablation Study

In this section, we conduct ablation studies using the VFIMamba-S model for efficiency.

**Effect of the S6 for VFI.** As a core contribution of this work, the S6 model balances computational efficiency and high performance for inter-frame modeling. To validate its effectiveness, as shown in Table 4, we experimented by removing the SSM model from the MSB (w/o SSM), replacing the MSB with convolutions from RIFE (Huang et al., 2022) (Convolution), or local inter-frame attention from EMA-VFI (Zhang et al., 2023) (Local Attention), or global inter-frame attention (Liu et al., 2024a) (Full Attention). We observed that only removing the S6 model resulted in a parameter reduction of only 0.7M but led to a significant performance drop across various datasets, underscoring the importance of S6. In comparisons with Convolution and Local Attention, we found that although the S6 model is relatively slower due to its multiple scanning directions, it achieves substantial performance improvements. Compared to Full Attention, S6 not only surpasses its performance but also offers faster inference speed and lower memory consumption. In summary, the S6 model indeed achieves a balance between computational efficiency and performance compared to existing models.

**Frame rearrangement for inter-frame modeling.** The rearrangement of input frames is crucial for inter-frame modeling using the S6 model. As analyzed in Section 3.3.2, we posit that interleaved rearrangement is more suitable for VFI tasks, and we provide experimental validation here. As shown in Table 5, we experimented with two different rearrangement methods in both horizontal and vertical scans. The results demonstrate that using interleaved rearrangement consistently achieves the best performance across all datasets, with significant improvements over other methods. These findings further validate our analysis that interleaved rearrangement offers superior spatiotemporal local modeling capabilities for VFI.

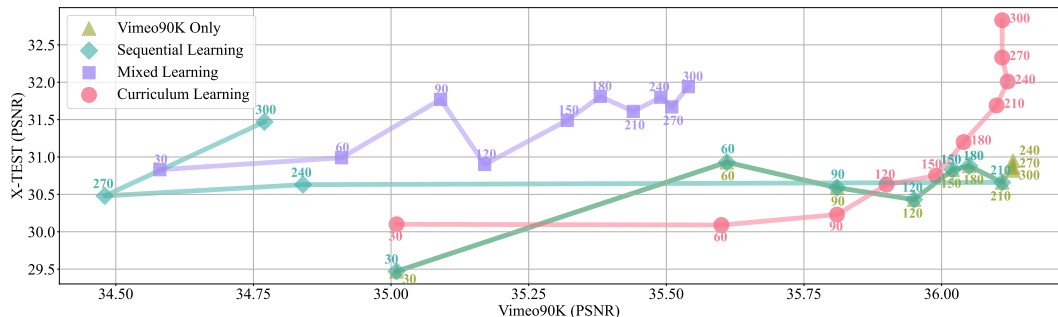

Figure 6: Performance of different learning methods, recorded every 30 epochs. Curriculum learning has the best performance in both the low-resolution and high-resolution benchmarks eventually.

Table 6: Performance of different methods without or with curriculum learning.

| | Curriculum Learning | Vimeo90K | X-TEST | | SNU-FILM | |
|---|---|---|---|---|---|---|
| | | | 2K | 4K | hard | extreme |
| RIFE | ✗ | 35.61/0.9797 | 31.10/0.8972 | 30.13/0.8927 | 30.36/0.9375 | 25.27/0.8601 |
| | ✓ | 35.60/0.9797 | 31.40/0.9142 | 30.23/0.9011 | 30.47/0.9376 | 25.38/0.8619 |
| EMA-VFI-S | ✗ | 36.07/0.9797 | 30.91/0.9000 | 29.91/0.8951 | 30.69/0.9375 | 25.47/0.8632 |
| | ✓ | 36.05/0.9797 | 31.15/0.9083 | 29.98/0.8988 | 30.73/0.9379 | 25.53/0.8652 |
| VFIMamba-S | ✗ | **36.13/0.9802** | 30.82/0.8997 | 29.87/0.8949 | 30.58/0.9378 | 25.30/0.8620 |
| | ✓ | 36.12/**0.9802** | **32.84/0.9328** | **31.73/0.9238** | **30.80/0.9381** | **25.59/0.8655** |

**Explore different learning strategy.** As described in Section 3.4, we proposed a curriculum learning strategy to fully harness the global modeling capabilities of the S6 model. In Figure 6, we present the performance of different learning strategies over training epochs on both Vimeo90K and X-TEST. In addition to the Vimeo90K Only and Sequential Learning strategies mentioned in Section 3.4, we also compared a baseline approach where the two datasets were directly mixed for training (Mixed Learning). The results indicate that as epochs increase, the Vimeo90K Only strategy improves performance exclusively on Vimeo90K with negligible change on X-TEST. Sequential Learning, while eventually enhancing X-TEST performance, significantly degrades performance on Vimeo90K. Mixed Learning shows a gradual increase in performance on both datasets but fails to achieve competitive results. Our proposed curriculum learning strategy, however, achieves the best performance on both datasets simultaneously by the end of training.

**Generalization of curriculum learning** To validate the generalization capability of curriculum learning, we also trained the RIFE (Huang et al., 2022) and EMA-VFI (Zhang et al., 2023) from scratch using curriculum learning. As shown in Table 6, after training, all models maintained their performance on the low-resolution dataset Vimeo90K while significantly improving performance on the X-TEST and SNU-FILM, fully verified the generalization of curriculum learning. Among these, our VFIMamba achieved the most significant improvement and the highest performance ceiling, further demonstrating the potential of the S6 model.

## 5 Conclusion

In this paper, we have introduced VFIMamba, the first approach to adapt the SSM model to the video frame interpolation task. To achieve global inter-frame modeling with linear complexity, we devise the Mixed-SSM Block (MSB) for efficient inter-frame modeling using S6. We also explore various rearrangement methods to convert two frames into a sequence, discovering that interleaved rearrangement is more suitable for VFI tasks. Additionally, we propose a curriculum learning strategy to further leverage the potential of the S6 model. Experimental results demonstrate that VFIMamba achieves the state-of-the-art performance across various datasets, in particular highlighting the potential of the SSM model for VFI tasks with high resolution.

## Acknowledgement

This work is supported by the National Key R&D Program of China (No. 2022ZD0160900), the National Natural Science Foundation of China (No. 62076119), the Fundamental Research Funds for the Central Universities (No. 020214380119), the Nanjing University- China Mobile Communications Group Co., Ltd. Joint Institute, and the Collaborative Innovation Center of Novel Software Technology and Industrialization.

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

# A Appendix

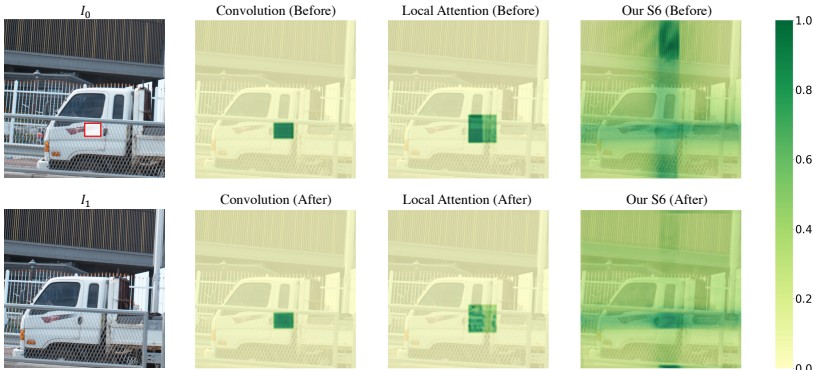

Figure 7: Visualizations of Effective Receptive Field (ERF) (Luo et al., 2016) of different models before and after training. We utilize the red area in $I_0$ to inspect its corresponding ERF in $I_1$. S6 model has a significantly larger receptive field and becomes more accurate after training.

## A.1 Broader impact

In this work, we introduce VFIMamba, the first video frame interpolation model based on SSMs. Video frame interpolation has wide-ranging applications in real-world video data processing, such as increasing the frame rate of AI-generated videos and generating slow-motion videos. Enhancing performance in various scenarios is crucial. However, as a research-oriented work, we trained our model on a very limited set of datasets (Vimeo90K (Xue et al., 2019) and X-TRAIN (Sim et al., 2021)), which might result in some degree of overfitting. Consequently, there could be significant artifacts when applied in real-world usage. This issue can be mitigated by training on a more diverse and extensive set of datasets.

## A.2 Limitations and future work

As the first work to explore the application of SSM models in frame interpolation tasks, we have achieved high performance, but there are still some limitations. First, although our method is much faster than attention-based methods, it still falls short of real-time requirements. Future work on designing a more efficient SSMs would be highly valuable. Second, in this work, we primarily focused on the role of SSM in inter-frame modeling and did not explore its use in the frame generation module. In the future, directly using SSM for generating intermediate frames could also be a promising direction for exploration.

## A.3 Visualizations on effective receptive field

To further evaluate the effective receptive field (ERF) of the S6 model in comparison with other efficient models (CNN, Local Attention) for inter-frame modeling, we used the method described by Luo et al. (2016). Given a specific region in $I_0$, we visualized the corresponding receptive fields in $I_1$ for different methods.

As shown in Figure 7, when the motion between $I_0$ and $I_1$ is significant, neither convolution nor local attention can focus on the corresponding region in $I_1$ before or after training. In contrast, the S6 model exhibits a larger global receptive field even before training, with notable concentration in both horizontal and vertical directions. We attribute this to the sequence rearrangement, where tokens closer together tend to have higher weights, a phenomenon also observed in VMamba (Liu et al., 2024b).

After training, the S6 model's focus becomes more concentrated on the horizontal region of $I_1$, aligning better with the specified region in $I_0$. This indicates that the S6 model can better capture dynamics even with significant motion between frames.

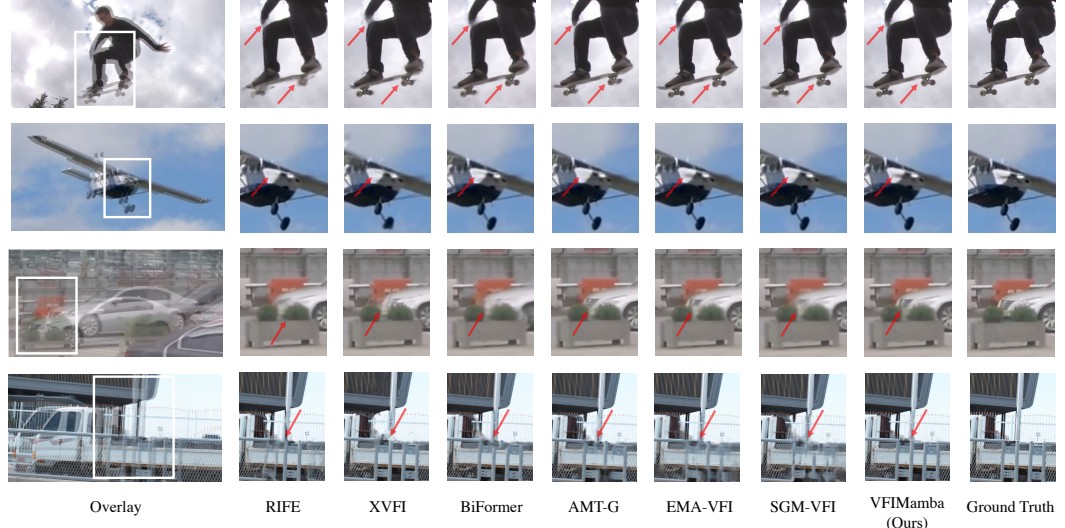

| Overlay | RIFE | XVFI | BiFormer | AMT-G | EMA-VFI | SGM-VFI | VFIMamba (Ours) | Ground Truth |

Figure 8: More Visualizations from SNU-FILM (Reda et al., 2022) and X-TEST (Sim et al., 2021).

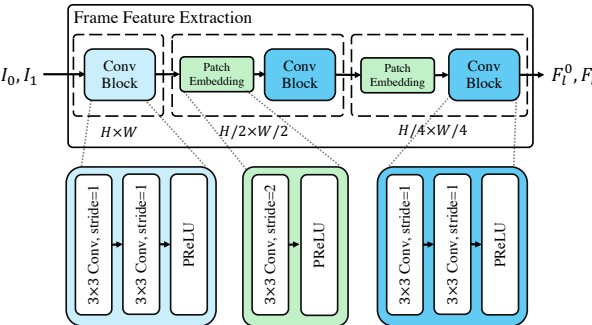

Figure 9: Details of frame feature extraction. The same color represents the same block structure.

## A.4 More qualitative comparison

As shown in Figure 8, we provide more visualization comparisons. VFIMamba demonstrates better visual quality compared to other methods.

## A.5 Model details

### A.5.1 Frame feature extraction

As shown in Figure 9, our frame feature extraction consists of multiple convolutional layers and PReLU (He et al., 2015). The first convolution maps the image from 3 channels to $C$, with $C = 16$ for VFIMamba-S and $C = 32$ for VFIMamba. Each time patch embedding is applied, the image resolution is halved, and the number of channels is doubled. Finally, we obtain the shallow features $F_l^i$ for each frame.

### A.5.2 Frame generation

As depicted in Figure 10, our frame generation includes an iterative intermediate flow estimation, local flow refinement, and appearance refinement using RefinNet. First, the intermediate flow estimation module uses the features $F_{ssm}^i$ obtained from inter-frame modeling with MSB for rough flow estimation. Specifically, we follow the design of EMA-VFI (Zhang et al., 2023), first utilizing features $F_{ssm}^i$ from the $\frac{H}{16} \times \frac{W}{16}$ scale and the original image $I_i$ for predicting the flow $f$ and occlusion

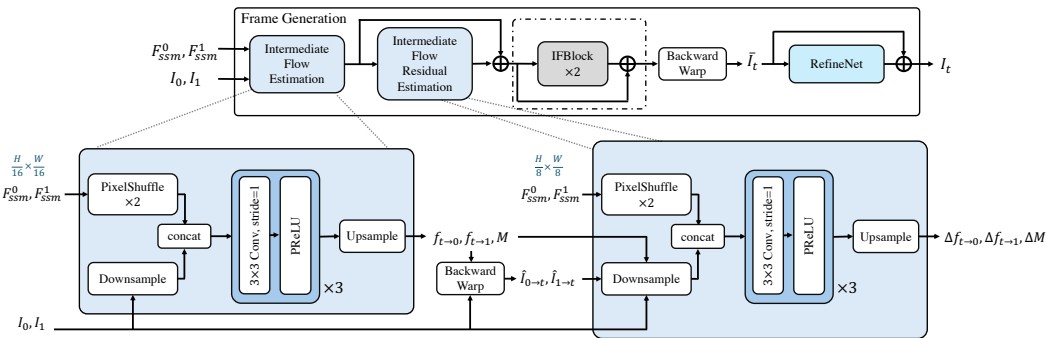

Figure 10: Details of frame generation. IFBlock is adopted from Huang et al. (2022), which is used optionally to enhance the local detail generation performance.

mask $M$ by several convolutional layers. Then, we iteratively estimate the flow residual $\Delta f$ and mask residual $\Delta M$ using the $F_{ssm}^i$ from the $\frac{H}{8} \times \frac{W}{8}$ scale. After that, inspired by Jia et al. (2022), which recognizes that the flow obtained through global inter-frame modeling may be coarse for high-resolution or large-motion scenes, we also introduce the IFBlock (Huang et al., 2022) to further enhance flow accuracy in local details. We then use the predicted motion to backward warp (Huang et al., 2022) the input frames to get the coarse intermediate frame $\bar{I}_t$. Finally, we adopt a U-Net-like (Ronneberger et al., 2015) structure to predict the appearance residual using shallow features $F_l^i$ and inter-frame features $F_{ssm}^i$, resulting in the final frame $I_t$.

## A.6 Training details

**Training loss** We used the same training loss as Zhang et al. (2023), which is a weighted combination of Laplacian loss (Niklaus & Liu, 2020) and warp loss (Liu et al., 2019), with weights of 1 and 0.5, respectively.

**Training setting** We used curriculum learning to train our model. For the data from Vimeo90K (Xue et al., 2019), we randomly cropped the frames from $256 \times 448$ to $256 \times 256$. For the data from X-TRAIN (Sim et al., 2021), since each sample contains 64 consecutive frames, we first randomly select two frames, starting with an interval of 1, which doubles every 50 epochs. Then, we randomly resized the frames from $512 \times 512$ to $S \times S$, where $S$ is initially 256 and increased by a factor of 1.1 every 50 epochs, and finally cropped them to $256 \times 256$ for alignment. The larger the resize size, the greater the motion magnitude of the generated data. The batch size for Vimeo90K is 32, and for X-TRAIN it is 8. We then applied time reversal and random rotation augmentations. We used AdamW as our optimizer with $\beta_1 = 0.9$, $\beta_2 = 0.999$, and a weight decay of $1 \times 10^{-4}$. With warmup for 2,000 steps, the learning rate was gradually increased to $2 \times 10^{-4}$, and then we used cosine annealing for 300 epochs to reduce the learning rate from $2 \times 10^{-4}$ to $2 \times 10^{-5}$. Following Jia et al. (2022); Park et al. (2023); Liu et al. (2024a), we also trained the IFBlock separately on Vimeo90K for 100 epochs with same training setting to further improve the accuracy of local optical flow at high resolutions. The same procedure is followed for all ablation experiments.

**Training time** VFIMamba and VFIMamba-S were both trained on 4 NVIDIA 32GB V100 GPUs. Training VFIMamba-S takes about 38 hours, while training VFIMamba takes about 108 hours.

## A.7 Evaluation protocols

In our paper, we primarily evaluated our methods on six benchmarks in terms of PSNR/SSIM(Wang et al., 2004): Vimeo90K (Xue et al., 2019), UCF101 Soomro et al. (2012), SNU-FILM (Choi et al., 2020), Xiph (Montgomery, 1994), X-TEST (Sim et al., 2021), and X-TEST-L (Liu et al., 2024a).

We followed the test procedures of Huang et al. (2022) for Vimeo90K and UCF101, Kong et al. (2022) for SNU-FILM, Niklaus & Liu (2020) for Xiph, Reda et al. (2022) for X-TEST with iterative 8× frame interpolation, and Liu et al. (2024a) for X-TEST-L with largest interval interpolation.

Table 7: Licenses and URLs for every benchmark, code, and pretrained models used in this paper.

| Assets | | License | URL |
|---|---|---|---|
| Benchmarks | Vimeo90K | MIT license | https://github.com/anchen1011/toflow |
| | UCF101 | non-commercial research and educational purposes | https://github.com/lxx1991/pytorch-voxel-flow |
| | SNU-FILM | MIT license | https://github.com/myungsub/CAIN |
| | XTEST (-L) | for research and education only | https://github.com/JihyongOh/XVFI |
| | Xiph | freely redistributable | https://media.xiph.org/video/derf/ |
| Codes and Pretrained Models | XVFI | for research and education only | https://github.com/JihyongOh/XVFI |
| | FILM | Apache-2.0 license | https://github.com/google-research/frame-interpolation |
| | RIFE | MIT license | https://github.com/hzwer/ECCV2022-RIFE |
| | IFRNet | MIT license | https://github.com/ltkong218/IFRNet |
| | BiFormer | Apache-2.0 license | https://github.com/JunHeum/BiFormer |
| | EMA-VFI | Apache-2.0 license | https://github.com/MCG-NJU/EMA-VFI |
| | AMT | CC BY-NC 4.0 | https://github.com/MCG-NKU/AMT?tab=License-1-ov-file |
| | SGM-VFI | Apache-2.0 license | https://github.com/MCG-NJU/SGM-VFI |

## A.8 License of datasets and pre-trained models

All the dataset we used in the paper are commonly used datasets for academic purpose. All the licenses of the used benchmark, codes, and pretrained models are listed in Table 7.

