# OpenReview forum: "VFIMamba: Video Frame Interpolation with State Space Models"
_NeurIPS.cc/2024/Conference — NeurIPS 2024 poster_

### Official Review · Reviewer_YfRu · 2024-07-10

**Soundness:** 3
**Presentation:** 3
**Contribution:** 3
**Rating:** 7
**Confidence:** 5

**Summary:**

Based on the popular S6 model's advantages of linear computational complexity and data-independent modelling capability, this paper applies it to VFI. Specifically, this paper proposes a token rearrangement strategy to learn the information of adjust frames, in addition to introducing a curriculum learning strategy to dynamically learn various motion magnitudes between adjust frames through joint training of vimeo90K and X-TRAIN. The model achieves the highest performance on existing commonly used VFI datasets.

**Strengths:**

1. This paper is the first to adapt the S6 model to the VFI.

2. Experimental results show that the proposed VFIMamba achieve performance while using competitive FLOPs.

**Weaknesses:**

1. It is recommended that the authors can demonstrate the accuracy of the interpolation results more intuitively by visualizing the error maps.

2. The experiments on table 4 are to demonstrate the computational validity of S6 model, and it is necessary to provide flops for the different models.

3. In limitations, the author says that the VFIMamba has faster speed, I would like to know the comparison of the runtime in table 2.

4. Lack of citations essential to the field, such as VFIT, TTVFI, ABME, EDSC, etc.

**Questions:**

I would like to see more analysis on the model effectiveness, such as the runtimes in Table 2.

**Limitations:**

The authors discuss methodological limitations in the supplementary material.

---

> ### Author Rebuttal · Authors · 2024-08-03
>
> We are grateful for your recognition and feedback on our work. We would like to respond to your concerns as follows:
>
> **Q.1** Visualization of error maps
>
> **R.1** Thank you for your suggestion. Visualizing the error maps can indeed provide a more intuitive demonstration of the accuracy of the interpolation results. We have provided an illustrative example in Figure 11 of the global response PDF, where VFIMamba shows clear advantages over other methods. We will add more comparative examples in the final version.
>
> **Q.2** FLOPs of Table 4
>
> **R.2** Thank you for your suggestion. The FLOPs of the different models in Table 4 are reported in the table below. We will include this in the final version.
> | Model | FLOPs(T) |
> | --- | --- |
> | w/o S6 | 0.23 |
> | Convolution | 0.27 |
> | Local Attention | 0.23 |
> | Full Attention | 0.59 |
> | S6 | 0.24 |
>
> **Q.3** Runtime comparison
>
> **R.3** Thank you for your suggestion. We have provided the runtime comparison in Table 2 of the global response PDF, and we welcome you to check that. We would also like to kindly remind you that in the limitations, we stated that VFIMamba is faster compared to the attention methods like SGM, with a runtime of 311ms vs. 942ms for 1024x1024 inputs. **Our primary goal was to achieve high-performance video frame interpolation while maintain efficient processing, rather than solely pursuing the fastest runtime.** We have discussed in detail how to further improve the runtime of our method in the response to **Reviewer Bwpp's Q.1**.
>
> **Q.4** Additional citations
>
> **R.4** Thank you for the reminder and these works are indeed important for the VFI task. We will add these citations in the final version.

---

> > ### Comment · Reviewer_YfRu · 2024-08-10
> >
> > Thanks to the author's reply, I raise my score to Accept. Looking forward to the author's future work to further address on accelerating SSM.

---

> > > ### Author Response · Authors · 2024-08-12
> > >
> > > We will continue to make effort to enhance the efficiency of our model in the future. Thank you so much for your kind recognition of our work.

---

### Official Review · Reviewer_9Hn5 · 2024-07-11

**Soundness:** 4
**Presentation:** 3
**Contribution:** 3
**Rating:** 7
**Confidence:** 5

**Summary:**

This paper introduces a novel video frame interpolation (VFI) method called VFIMamba. VFIMamba is the first method that combines the State-Space Model Mamba with VFI architectures and therefore, it has the advantage of a linearly growing complexity w.r.t. the resolution while maintaining the ability to utilize global receptive fields similar to vision transformers. In order to apply the idea to VFI, architectural modifications have been proposed to handle 2 frames as input. Further, a novel curriculum learning strategy is used to increase the models performance across a large range of motions. The method has been evaluated on multiple datasets and w.r.t. to various other methods, achieving state-of-the-art PSNR values, especially improving the performance for high-resolution frames (2K and 4K).

**Strengths:**

- Combination of an emerging alternative to transformers (MAMBA) with VFI methods that allow for higher resolution frame interpolation due to linear complexity growth. The paper is interesting to read.
 - Discussion and evaluation of different sequence arrangements.
- Introduction of a relatively simple curriculum learning strategy for VFI with experiments confirming that this strategy is beneficial for VFI methods in general when having to deal with large motions.
- Reaches new state-of-the-art performance
- Exhaustive ablation studies proving the effectiveness for each of their introduced modules
- Supplement contains a video with qualitative comparisons, although only a few short sequences and given the coarse time steps, it is difficult to clearly judge the temporal consistency.

**Weaknesses:**

- It is unclear how frames at arbitrary time steps are computed to perform 8x interpolation in table 3. It should also be directly clear from the caption of table 3, that 8x interpolation is evaluated (these details are only in the appendix).
- In general, a lot of important information is in the appendix. It is helpful to add at least a reference from the main paper to the appendix that there is more information, e.g., such as the evaluation and the experiment on generlalization of curriculum learning.
- Unclear how the FLOPs requirement scales with resolution. A plot/table showing the FLOPs compared to the input resolution for VFIMamba and other methods would be interesting to get a better feeling for the scaling w.r.t. frame resolution and a comparison of memory footprints might be interesting.
- Ideally, it should be mentioned in table 2 and 3 on which dataset the other methods have been trained on, especially for methods where the original paper proposed multiple versions such as XVFI.
- Some recent methods especially for high-resolution data are missing in Table 3 such as [A] and [B]. Especially [A] has only been included in Tab. 2 although their focus is also on high-resolution datasets and code for X-Test is directly available: https://github.com/feinanshan/M2M_VFI/blob/main/Test/bench_xtest.py


[A] Hu et al. Many-to-many Splatting for Efficient Video Frame Interpolation, 2022

[B] Nottebaum et al., Efficient Feature Extraction for High-resolution Video Frame Interpolation, 2022.
It would be nice to have longer sequences, and playing them a bit more fluently to get a feeling for the temporal and visual consistency.

Minor:
- Some colors difficult to see in overlay in Figure 1.
- L. 126, Check sentence “where contains”

**Questions:**

- Table 3: There is a discrepancy between many of the reported number in their respective original papers and in this paper on X-TEST for 2k and 4k, e.g.,   EMA-VFI-S reported 30.89dB instead of 29.91dB and BiFormer reported 31.32dB instead 31.18. Based on the experience of the reviewer it is possible to reproduce these numbers using the correct evaluation protocol.

- How is 8x interpolation performed?

- Unclear why there is no FLOPs measurement and no evaluation of SNU-FILM for SoftSplat given that the source code + trained models are publicly available (https://github.com/sniklaus/softmax-splatting)

**Limitations:**

Discussion of limitations is only in the appendix. It would be better to discuss them already in the main paper.

Additionaly, has the method similar limitations as MAMBA (the base model for this entire work). Therefore, the compute requirements are still relatively high even though they do not have to compute an attention matrix anymore.

---

> ### Author Rebuttal · Authors · 2024-08-03
>
> Thank you for your positive and constructive suggestions. We have the following responses to your concerns:
>
> **Q.1** *How is 8x interpolation performed?*
>
> **R.1** As mentioned in line 497, we followed the testing procedure of FILM [1] and used an iterative approach for frame interpolation. Specifically, we first generated an intermediate frame based on the input two frames, and then, using a divide-and-conquer strategy, we further divided the first frame and the generated intermediate frame, as well as the generated intermediate frame and the last frame, to iteratively generate the remaining frames. Thank you for the reminder, and we will further emphasize the specific testing procedure in the main text.
>
> **Q.2** *A lot of important information is in the appendix*
>
> **R.2** We sincerely appreciate your suggestion and will incorporate the mentioned content into the main text in the final version, with more references to the appendix.
>
> **Q.3** *How the FLOPs/memory scales with resolution*
>
> **R.3** We have included the comparison of FLOPs and memory usage of different methods at various resolutions in the global response PDF. As shown in Figure 10, our method has a clear advantage in FLOPs and memory usage at high resolutions.
>
> **Q.4** *It should be mentioned in Table 2 and 3 on which dataset the other methods have been trained on*
>
> **R.4** Thank you for your suggestion. We have updated the information in Tables 2 and 3 in the global response PDF to include the specific training datasets for each method. We will also add this information in the final version.
>
> **Q.5** *Some recent methods especially for high-resolution data are missing in Table 3*
>
> **R.5** Thank you for the reminder. We have added the results of these two papers under the same testing procedure in Table 3 of the global response PDF, and we will also include them in the final version.
>
> **Q.6** *A discrepancy between many of the reported numbers in their respective original papers and in this paper on X-TEST for 2k and 4k*
>
> **R.6** First, we would like to respectfully remind you that the results in different papers may have used different quantization methods (whether to round the network output) and test functions (the SSIM function in sklearn and the MATLAB-style SSIM can produce different results). **To make a fair comparison, we have re-tested all the open-source models under the same testing procedure, which may lead to minor differences in performance compared to their original results**. As for EMA-VFI-S, it used a model trained on the Vimeo90K **septuplet** dataset for testing X-TEST, while our method and all the other methods used models trained on the **triplet** dataset for testing. **For the fair comparison, we tested the EMA-VFI-S model trained on the triplet dataset** using our iterative frame interpolation method as in **Q.1**, resulting in a performance of 29.91 dB instead of 30.89 dB. Regarding the performance of BiFormer, BiFormer did not publicly release its test code and we used the test code of SGM [2]. The results were consistent with the SGM, so we initially thought we had successfully reproduced the correct results. Unfortunately, after careful examination and comparison,  we identified an inconsistency between the input processing of SGM's testing procedure and BiFormer's training procedure, making the results of BiFormer lower than it should be.  After re-conducting the proper testing, we also obtained a result of 31.32 dB, and we have also updated the results of BiFormer on other datasets in Table 3 of the global response PDF. **We also rechecked all the results in Table 3 and ensured their accuracy.** We are truly grateful for your reminder, and we will also remind the authors of SGM to correct the BiFormer results.
>
> **Q.7** *Performance of SoftSplat*
>
> **R.7** Thank you for the reminder. We sincerely apologize for not finding that SoftSplat has already been open-sourced. The performance of Softsplat under the same testing procedure has been updated in Table 2 of the global response PDF, and you are welcome to check it.
>
> **Q.8** *Minor errors*
>
> **R.8** We will further improve the color distinction in Figure 1 and carefully check the grammatical details of the paper. Thank you for the reminder.
>
> > [1] Reda, Fitsum, et al. "Film: Frame interpolation for large motion." European Conference on Computer Vision. Cham: Springer Nature Switzerland, 2022.
>
> > [2] Liu, Chunxu, et al. "Sparse Global Matching for Video Frame Interpolation with Large Motion." Proceedings of the IEEE/CVF Conference on Computer Vision and Pattern Recognition. 2024.

---

> > ### Comment · Reviewer_9Hn5 · 2024-08-12
> >
> > I thank the authors for the answers, and highly appreciate the extensive additional results and verification of the numbers. Minor detail, it would be nice if in the final version the FLOPs and runtime will also be added for 2K/4K. The additional provided details on training and additional ablations provided in the rebuttal pdf confirm the contribution of the paper and I still recommend the paper for acceptance.

---

> > > ### Author Response · Authors · 2024-08-12
> > >
> > > We will be sure to further polish the final version based on your helpful suggestions. Thank you so much for your kind recognition of our work.

---

### Official Review · Reviewer_Bwpp · 2024-07-13

**Soundness:** 3
**Presentation:** 3
**Contribution:** 3
**Rating:** 7
**Confidence:** 4

**Summary:**

The paper presents a novel approach for video frame interpolation using Selective State Space Models (S6). The authors introduce VFIMamba, a method designed to efficiently and dynamically model inter-frame information. This method features the Mixed-SSM Block (MSB), which rearranges tokens from adjacent frames in an interleaved manner and applies multi-directional S6 modeling. Additionally, the paper proposes a curriculum learning strategy to progressively improve the model's ability to handle varying motion magnitudes. Experimental results show that VFIMamba achieves state-of-the-art performance on various benchmarks, especially in high-resolution scenarios.

**Strengths:**

Originality: The introduction of the S6 model into video frame interpolation tasks is a novel contribution. The use of Mixed-SSM Blocks and the interleaved token arrangement are creative solutions to enhance inter-frame modeling.
Quality: The paper presents thorough experiments and comparisons with state-of-the-art methods. The quantitative results demonstrate significant improvements in performance, particularly in high-resolution and large-motion scenarios.
Clarity: The methodology is clearly explained, with detailed descriptions of the proposed model components and training strategies. The visualizations and tables effectively support the claims made in the paper.
Significance: The VFIMamba method addresses key challenges in video frame interpolation, such as the need for large receptive fields and efficient computation.

**Weaknesses:**

Real-time Application: Although VFIMamba achieves high performance, it still falls short of real-time requirements. The paper could benefit from a discussion on potential strategies to improve inference speed.
Performance Gap Analysis: On low-resolution datasets, VFIMamba-S fails to yield second-best scores on most benchmarks and only outperform baselines with comparable FLOPs (e.g., EMA-VFI-S) by a relatively small margin.

**Questions:**

1. What specific optimizations could be applied to VFIMamba to make it suitable for real-time applications? Are there trade-offs between speed and accuracy that need to be considered?
2. Could the authors elaborate on the specific factors contributing to the performance gap, and suggest potential directions to address these issues?

**Limitations:**

The authors have acknowledged several limitations of their work, including the resource-intensive nature of training VFIMamba and the current inability to meet real-time requirements.
The authors suggest future work on designing more efficient SSMs and exploring the application of SSMs in the frame generation module.

---

> ### Author Rebuttal · Authors · 2024-08-03
>
> We sincerely thank you for the recognition and suggestions regarding our work, and our responses to your questions are as follows:
>
> **Q.1** *What specific optimizations could be applied to VFIMamba to make it suitable for real-time applications? Are there trade-offs between speed and accuracy that need to be considered?*
>
> **R.1** Thank you for this insightful question. The running speed of SSMs is mainly related to two aspects: the length of the input sequence and the number of scans performed on the sequence. Regarding the sequence length, we can reduce the number of tokens by fusing the neighboring tokens in the spatio-temporal position (e.g., through ToMe [1]). However, this may also have an impact on the performance of fine-grained spatio-temporal modeling. As for the number of scans, we currently follow VSSM [2] and use 4 scan directions for modeling. In the future, we can reduce the number of scans by dynamically selecting the necessary scan directions for video frame interpolation.
>
> **Q.2** *Could the authors elaborate on the specific factors contributing to the performance gap, and suggest potential directions to address these issues?*
>
> **R.2** Thank you for your question. First, we would like to kindly remind you that except for EMA-VFI-S, VFIMamba-S has a significant performance advantage over other models with similar FLOPS (AdaCof, XVFI, M2M, RIFE, etc.). As for why the performance gap with EMA-VFI-S cannot be widened, EMA-VFI-S belongs to the local attention-based method, and as shown in Table 1 of the paper, the biggest advantage of Mamba over local attention-based methods is the global receptive field, which may not have a significant impact in low-resolution cases. In the future, perhaps combining SSMs with some more fine-grained local modeling methods can further improve the performance at low resolutions.
>
> > [1] Bolya, Daniel, et al. "Token merging: Your vit but faster." arXiv preprint arXiv:2210.09461 (2022).
>
> > [2] Liu, Yue et al. “VMamba: Visual State Space Model.” ArXiv abs/2401.10166 (2024).

---

> > ### Comment · Reviewer_Bwpp · 2024-08-12
> >
> > Thanks to the authors' reply, I raise my score to Accept since I am satisfied with the authors' response regarding performance gap. I am looking forward to the author's future work to further address on accelerating SSM.

---

> > > ### Author Response · Authors · 2024-08-12
> > >
> > > We are truly grateful for your kind recognition of our work.

---

### Official Review · Reviewer_A9TL · 2024-07-21

**Soundness:** 3
**Presentation:** 3
**Contribution:** 2
**Rating:** 6
**Confidence:** 4

**Summary:**

The paper introduces Mamba-based video frame interpolation. To fully incorporate the power of Mamba, the paper proposes an interleaving rearrangement method. Using this method, the SSM scans the same location tokens of 2 frames together instead of processing each frame separately. The paper also proposes curriculum learning with large motion data. The experiments show the improvement of using these ideas.

**Strengths:**

This paper includes three contributions: 1) Use of Mamba for image interpolation, 2) Interleaved rearrangement, and 3) curriculum learning.
- The idea is simple but effective for video frame interpolation. All these contributions are justified throughout the paper (especially Sections 3 and 4).
- The paper is easy to follow.

**Weaknesses:**

I have comments and questions about the final comparisons (Tables 2 and 3).
>Most methods training models exclusively on the Vimeo90K (Xue et al., 2019). .... (2) Sequential Learning: To mitigate the limitations of training solely on Vimeo90K, some methods (Liu et al., 2024a; Park et al., 2023) further train the model on X-TRAIN (Sim et al., 2021) after initial training on Vimeo90K.

- Based on the description above, the proposed model used both Vimeo90K and X-TRAIN for training, but not all methods use both datasets. It's unclear which models are trained only on Vimeo90K or Vimeo90K+X-TRAIN. The improvement can be due to the increase of training datasets especially since the improvement is marginal compared to some models such as EMA-VFI (for low resolution), AMT-L, AMT-G, and SGM-VFI. I understand that curriculum learning improves performance compared to the simple data mix, but the current tables are not comparable if the trained datasets are not the same across the models.
    - I ask authors to add which models are trained only on Vimeo90K or Vimeo90K+X-TRAIN in the tables.
    - If some comparable models are trained only on Vimeo90K, can authors add the comparisons of these models with Vimeo90K+X-TRAIN? I'm not sure if it's possible during the rebuttal period.

- As some comparing models are better or similar to the proposed model, it's difficult to compare which ones are better. Adding the average of each model on the right column would be helpful.

- How can the interleave rearrangement be efficiently implemented? It would be good to add details of how it is implemented in the paper. Also, is there any speed issue of using the interleave rearrangement instead of the sequential one?

**Questions:**

See the weaknesses.

**Limitations:**

The limitation is discussed in the paper and is reasonable.

---

> ### Author Rebuttal · Authors · 2024-08-03
>
> We sincerely appreciate your feedback on the experimental and implementation details of our work. We would like to provide the following responses:
>
> **Q.1** *Regarding which models are trained only on Vimeo90K or Vimeo90K+X-TRAIN in the tables.*
>
> **R.1** Thank you for your suggestion. We would like to kindly remind you that the recent published papers (e.g., BiFormer [1] and SGM [2]) also did not specify the exact training dataset used for the compared methods. We have followed their settings, but **we agree that adding this detail will make the comparison more informative for the readers**. We will add the relevant information to Tables 2 and 3 as shown in the PDF of the global response.
>
> **Q.2** *Add the comparisons of these Vimeo90K-only models trained on Vimeo90K+X-TRAIN.*
>
> **R.2** As discussed in Table 6 and Appendix A.4 of the paper, we have already retrained the RIFE [3] and EMA-VFI-S [4] models on Vimeo90K+X-TRAIN. The results show that through curriculum learning, these two methods have achieved performance improvements on the high-resolution dataset, while maintaining their performance on Vimeo90K, demonstrating the generalization of our proposed curriculum learning training strategy. Meanwhile, **VFIMamba still exhibits the best performance under the same training setting**, indicating the higher upper bound of the VFIMamba structure.
>
> **Q.3** *Add the average of each model on the right column.*
>
> **R.3** Thanks for your suggestion. We will add the relevant information to Tables 2 and 3 as shown in the PDF of the global response.
>
> **Q.4** *Regarding the implementation of the interleave rearrangement.*
>
> **R.4** The implementation of the interleave rearrangement is very simple and efficient, which can be achieved through the following few lines of PyTorch code:
>
> ```python
> def interleave_merge(self, x1, x2):
>     # x1 is the first frame, x2 is the second
>     B, C, H, W = x1.shape
>     N = H * W
>     # Flatten the 2D tokens and transpose to B x N x C format
>     x1 = x1.view(B, C, -1).transpose(1, 2)
>     x2 = x2.view(B, C, -1).transpose(1, 2)
>     # Concatenate in the C dimension to get B x N x 2C, and then reshape to get B x 2N x C, and PyTorch will automatically interleave the tokens of the two frames
>     x = torch.cat([x1, x2], dim=-1).reshape(B, 2*N, C)
>     # Reshape back to B x C x 2N
>     return x.transpose(1, 2).contiguous()
> ```
> Both the interleave rearrangement and sequential rearrangement only involve reshaping and rearranging the token, which are linear-time operations. Therefore, the running efficiency of interleave rearrangement and sequential rearrangement can be considered approximately the same.
>
> > [1]Park, Junheum, Jintae Kim, and Chang-Su Kim. "Biformer: Learning bilateral motion estimation via bilateral transformer for 4k video frame interpolation." Proceedings of the IEEE/CVF Conference on Computer Vision and Pattern Recognition. 2023.
>
> > [2]Liu, Chunxu, et al. "Sparse Global Matching for Video Frame Interpolation with Large Motion." Proceedings of the IEEE/CVF Conference on Computer Vision and Pattern Recognition. 2024.
>
> > [3]Huang, Zhewei, et al. "Real-time intermediate flow estimation for video frame interpolation." European Conference on Computer Vision. Cham: Springer Nature Switzerland, 2022.
>
> > [4]Zhang, Guozhen, et al. "Extracting motion and appearance via inter-frame attention for efficient video frame interpolation." Proceedings of the IEEE/CVF Conference on Computer Vision and Pattern Recognition. 2023.

---

> > ### Comment · Reviewer_A9TL · 2024-08-12
> > **response to the rebuttal**
> >
> > Thank you for the answers.
> > My major concern was the unfair comparisons in Tables 2 and 3 as the paper uses an extra dataset. The authors' answers and additional information in Tables 2 and 3 (also Table 6) resolve my concern. Please add these details to the revised version. Also, please move Table 6 to the main paper; it's an important ablation.
> > As the paper includes clear contributions and improvements, I increase my rating.

---

> > > ### Author Response · Authors · 2024-08-12
> > >
> > > We will be sure to incorporate your suggestions into the final version. Thank you so much for your kind acknowledgement of our work.

---

### Author Rebuttal · Authors · 2024-08-03

We sincerely appreciate all reviewers' efforts in reviewing our paper and giving insightful comments as well as valuable suggestions. We are glad to find that the reviewers generally acknowledge the following contributions of our work.

* **Framework.** We are the first to adapt the S6 model to the VFI task [YfRu,A9TL,Bwpp,9Hn5] for addressing key challenges in video frame interpolation, such as the need for large receptive fields and efficient computation. The use of Mixed-SSM Blocks and the interleaved token arrangement are effective solutions for enhancing inter-frame modeling [A9TL,Bwpp,9Hn5].
* **Training strategy.** We propose a novel curriculum learning strategy that is beneficial for VFI methods when having to deal with large motions [A9TL, 9Hn5].
* **Experiments.** Our experimental results achieve state-of-the-art performance on most datasets, particularly in high-resolution and large-motion scenarios [Bwpp,9Hn5,YfRu]. Furthermore, we have conducted exhaustive ablation studies to demonstrate the effectiveness of our proposed method [A9TL, Bwpp, 9Hn5].

As suggested by the reviewers, we include the following contents in the revised manuscript to further polish our paper. The major revision is summarized as follows. Our detailed responses can be found in each response section to the reviewers.

* **Adding more detailed information to the main comparison Table 2 and Table 3.** This includes: *1)* The training dataset for each method [A9TL, 9Hn5]. *2)* The average performance of each model [A9TL]. *3)* Results for M2M, FLDR, BiFormer and SoftSplat [9Hn5]. *4)* Runtime [YfRu]. We have provided the updated Table 2 and Table 3 in the PDF below.
* **More visualization comparisons.** This includes visualizations on how the FLOPs/memory requirement scales with resolution [9Hn5], as well as error maps [YfRu]. We have provided these in the PDF below.
* **Transferring important information from the appendix to the main paper**. This includes detailed testing procedures and the experiment on the generalization of curriculum learning [A9TL, 9Hn5].

---

### Comment · Area_Chair_AG9d · 2024-08-09

Dear Reviewers,

Thank you very much again for your valuable service to the NeurIPS community.

As the authors have provided detailed responses, it would be great if you could check them and see if your concerns have been addressed. Your prompt feedback would provide an opportunity for the authors to offer additional clarifications if needed.

Best regards,

AC

---

### Decision · Program_Chairs · 2024-09-25

**Decision:**

Accept (poster)

**Comment:**

All the reviewers unanimously recognize the novelty/significance of combining a new architecture (MAMBA) and the VFI task, the thorough experimental evaluation, the significance of a relatively simple curriculum learning strategy for VFI, and strong state-of-the-art performance of the proposed approach. The AC thus recommends to accept this paper.

The authors are strongly encouraged to incorporate the new results into the final version of this paper.